# Reanimation of the Smile with Neuro-Vascular Anastomosed Gracilis Muscle: A Case Series

**DOI:** 10.3390/diagnostics12051282

**Published:** 2022-05-21

**Authors:** Helen Abing, Carina Pick, Tabea Steffens, Jenny Shachi Sharma, Jens Peter Klußmann, Maria Grosheva

**Affiliations:** Department of Otorhinolaryngology, Head and Neck Surgery, Faculty of Medicine, University of Cologne, 50937 Cologne, Germany; carina.pick@gmx.de.de (C.P.); tabea.steffens@t-online.de (T.S.); shachi.sharma@uk-koeln.de (J.S.S.);

**Keywords:** longstanding facial palsy, gracilis transfer, masseteric nerve, needle electromyography, outcome

## Abstract

Background: The aim of our manuscript was to evaluate the time course of clinical and electromyographical (EMG) reinnervation after the reanimation of the smile using a gracilis muscle transplant which is reinnervated with the masseteric nerve. Methods: We present a case series of five patients with a longstanding peripheral facial palsy, who underwent a reanimation of the lower face using a gracilis muscle transplant with masseteric nerve reinnervation from June 2019 to October 2020. Trial-specific follow-up examinations were carried out every three months using clinical assessment and EMG, up to 12 months after the surgery. The grading was carried out using the House–Brackmann scale (HB), the Stennert Index, and a self-designed Likert-like scale for graft reinnervation and smile excursion. Results: The surgery was feasible in all of the patients. The reanimation was performed under general anesthesia in an inpatient setting. Postoperative complications which resulted in prolonged hospitalization occurred in two of the five patients. All of the patients showed a preoperative flaccid facial palsy. The first single reinnervation potentials were detected 3.1 ± 0.1 months after surgery. After 5.6 (±1.4) months, in three (3/5) patients, clear reinnervation patterns were present. Clinically, the patients obtained symmetry of the face at rest after 5.6 (±1.4) months, and could spontaneously smile without the co-activation of the jaw after an average time of 10.8 (±1.8) months. All of the patients were able to express a spontaneous emotion-stimulated smile after one year. Conclusion: Micro-neurovascular gracilis muscle transfer reinnervated with a masseteric nerve is a sufficient and reliable rehabilitation technique for the lower face, and is performed as a single-stage surgery. The nerve supply via the masseteric nerve allows the very rapid and strong reinnervation of the graft, and results in a spontaneous smile within 10 months.

## 1. Introduction

Peripheral facial nerve palsy is the most common peripheral cranial nerve disorder. Its etiology is very diverse. Although the most common, idiopathic facial nerve palsy (FP) (Bell’s palsy, with 60–75% of all cases) heals in nearly 80–90% of patients, other etiologies are known to be associated with defective healing, or even to result in a longstanding flaccid paralysis [1]. FP dramatically limits the quality of life on various levels. Dysphagia, speech disorders, xerophthalmia and consecutive corneal damage are some but not all of the possible physical consequences of the FP. Furthermore, limited nonverbal communication and the change of the appearance of the face significantly impact daily life. As a consequence, patients often suffer from psychological stress and social isolation [2]. For both aspects, the proper function of the lower face, especially the existence of the spontaneous smile, is crucial [3]. 

Surgical nerve reconstruction and reconstructive surgery are considered to be the gold standard for the reanimation of longstanding FP [3]. However, as structurally irreversible changes occur in the facial musculature, facial nerve fibers and the synaptic connections within the first 2–3 years after the palsy’s onset, the reanimation of a longstanding paralysis requires complex surgical procedures [4]. For the reanimation of the smile, the transfer of the gracilis muscle, which then mimics the function of the zygomatic muscle, is internationally established. Previous studies have demonstrated the benefit of this procedure due to improvement of functional aspects, a positive effect on social interaction and self-acceptance, and an increase in quality of life [5]. Muscle transfer using a gracilis graft was first described by Harii et al. in 1976 [6]. Here, nerval reinnervation via cross-face transfer was considered the most advantageous [6]. The procedure is associated with good motor function, the almost symmetrical mobility of the face, and simultaneous control via the facial nerve of the opposite side [7,8]. However, this reanimation technique requires multi-stage surgery, and is associated with the partial sacrifice of the healthy facial nerve branches, which might result in synkinesis. 

The reinnervation of the gracilis graft with the masseteric nerve represents an alternative approach. The harvesting of the muscle graft and the reinnervation can be achieved during a single stage operation. The nerve is accessible in the already exposed surgical site [9,10]. Motor unit reinnervation is expected to be faster due to the shorter innervation distance compared to the facial nerve [11]. However, for a good functional outcome, the contraction of the gracilis graft has to be triggered by biting, which requires the sufficient compliance of the patient during rehabilitation as well as the sufficient plasticity of the brain [12,13,14]. In more than half of the cases, a spontaneous, emotionally stimulated smile can be obtained without the use of biting or chewing [15]. 

The aim of our manuscript was to describe the surgical procedure and the time-course of clinical and electromyographical reinnervation after the reanimation of the smile using a gracilis muscle transplant, which was reinnervated with the masseteric nerve.

## 2. Patients and Methods

We present a case series of 5 patients with a longstanding peripheral facial palsy, who underwent a reanimation of the lower face using a gracilis muscle transplant with masseteric nerve reinnervation from June 2019 to October 2020. We analyzed the time course of the reinnervation of the gracilis muscle and the final functional outcome after one year.

All of the patients signed a written consent form for data collection. Only adult patients with a longstanding facial nerve palsy were included. The study was approved by the Ethics Committee of the University of Cologne (approval Nr. 22-1106-retro, Cologne, Germany).

### 2.1. Surgery

All of the patients underwent the surgical procedure under general anesthesia in an inpatient setting. All of the operations were carried out by surgeons MG and SJS in a simultaneous two-team approach.

First, the dissection of the face and the parotid area was carried out (MG). The vessels including the retromandibular vein plexus and superficial temporal artery were identified after the preauricular incision. Then, the masseteric triangle—including the zygomatic bone, the buccal branch of the facial nerve, and the mandibular joint—was exposed [16,17]. The masseteric nerve was identified and dissected to the maximum extent. For the positioning of the gracilis graft, the subcutaneous tissue of the cheek was mobilized to the nasolabial fold, and further to the oral commissure. The dissection was carried out in a sub-SMAS deep plain. During the subcutaneous dissection of the lower and upper lip, the orbicularis oris muscle (OOM) was identified. In order to enable the tissue dissection and future positioning of the gracilis muscle, skin incisions along the nasolabial fold and submentally were carried out.

The second surgical team (SJS) harvested the gracilis muscle together with the neurovascular pedicle, including the obturator nerve and a branch of the medial circumflex femoral artery or of the profunda femoris artery. The muscle was removed in its entire width.

For the positioning of the graft, the inferior part of the muscle was separated into two equal pedicles. The superior part of the muscle included the recipient vessels and the obturator nerve (Figure 1). First, the microsurgical end-to-end suture of the masseteric and obturator nerve was carried out using non-resorbable Ethilon 9-0 (Ethicon^®^, Johnson & Johnson Medical GmbH, Norderstedt, Germany) sutures. The donor artery (the superficial temporal artery) was sutured end-to-end to the recipient artery using Ethilon 9-0 sutures. For the venous anastomosis (retromandibular vein), a coupler system (Gem Coupler^®^, Baxter Healthcare, Opfikon, Switzerland) was used (Figure 2). In one patient, we chose the facial artery and vein as donor vessels. The muscle graft was positioned in such way that the peripheral pedicles reached the OOM medially. The muscle pedicles were sutured to the OOM fibers of the upper and lower lip with non-resorbable sutures (Ethibond 4-0, Ethicon^®^). The united pedicles were also sutured to the OOM in the oral commissure. We used a micro-osteosynthesis 5- to 4-hole plate (KLS Martin Group) to attach the superior part of the muscle graft to the zygomatic bone. A wound drain was placed in both surgical areas. Wound closure was performed in layers using subcutaneous and skin sutures. 

In some patients, additional static reanimation procedures of the upper face were carried out during the same operation. These were forehead and brow lifts, eyelid loading, and lateral canthopexy.

### 2.2. Data Assessment and Follow-Up

Data on the patients’ characteristics, the surgery duration in minutes, the duration of the hospitalization, and data on postoperative complications were assessed. 

Trial-specific follow-up examinations were carried out every three to four months. Facial function was documented preoperatively and during the follow-up visits. The grading was carried out using the House–Brackmann scale (HB) and the Stennert Index [18]. The Stennert Index allows the grading of the resting face (0 for regular function to 4 for flaccid complete paralysis) and during voluntary movement (0 for no paresis and 6 for complete flaccid paralysis) [18]. As neither scale (the Stennert Index nor the HB Scale) was originally designed to evaluate facial symmetry after reconstruction, we used the Likert-like scale to evaluate the clinical outcome. Clinical outcomes were documented as follows: asymmetry at rest, flaccid paralysis; asymmetry at rest, minimal improvement of the muscle tone while biting; improved muscle tone and symmetry at rest while biting; symmetry at rest while biting, spontaneous smile possible; symmetry at rest, spontaneous smile without biting. 

Needle electromyography (EMG) was carried out during each follow-up examination until the primary outcome (the clear excursion of the smile) was clearly visible (Nicolet Electromygoraphy^®^, Natus Medical, San Carlos, CA, USA). The EMG results were rated as follows: preoperative EMG, no activity; single reinnervation potentials while biting; single reinnervation potentials during smiling without biting; clear reinnervation pattern without biting; clear clinical reinnervation, no EMG examination necessary.

### 2.3. Statistical Analysis

The statistical analysis was performed using SPSS software, version 28.0 (IBM Corporation, Armonk, New York, NY, USA). We present the quantitative variables as the mean ± SD, and the qualitative variables as absolute numbers and percent values. 

## 3. Results

### 3.1. Characteristics of the Patients and Surgery

From June 2019 to October 2020, five patients underwent the reanimation procedure. The characteristics of the patients are shown in Table 1. The patients, three of them female, were 48.7 ± 16.1 years old. The median duration of paralysis was 18.9 years (range 2–55 years). The paralysis was on the right side of the face in all of the patients. Three patients showed complete flaccid paralysis before the surgery. One patient showed a paralysis score of HB IV, for a good resting tone of the facial muscles. Another patient consulted us because of congenital middle face dysplasia and showed a facial palsy with an HB score of V, with preserved function of the frontalis muscle. Preoperative needle EMG showed no voluntary activity of the nasalis and zygomatic muscles, nor of the depressor anguli oris muscle (DAOM) in any of the patients.

The surgical procedure, as described above, was feasible in all of the patients. The mean duration of the surgery was 506 (±62.5) minutes, and the mean duration of the hospitalization was 11 (±3.7) days. Two patients were hospitalized longer because of a postoperative complication (patient 1: massive hemorrhage into the thigh because of derailed anticoagulation; patient 3: delayed wound healing). Patient 5 had prolonged postoperative ventilation due to a complex dysplasia of the middle and lower face.

For vascular anastomosis, we chose the superficial temporal artery and retromandibular vein as donor vessels in four cases because of the good accessibility in the already-opened surgical site. In patient 1, these vessels were not preserved during previous surgery for parotid cancer. For this reason, we chose the facial artery and vein as donors. 

### 3.2. Follow-Up and Outcome

The follow-up took place every 3.1 (±1.1) months, on average. The first follow-up took place after 3.1 (±1.0) months, and the last follow-up took place after 12.3 (±3.0) months. The timeline of the visits is displayed in Table 2. 

The functional outcomes and EMG results are summarized in Figure 3 and Figure 4, respectively. The first improvement of the musculature’s tone was detected at the first follow-up, after an average time of 3.1 (±1.0) months. At this stage, asymmetry at rest was still evident, and the masseter muscle was needed to increase the muscle tone by biting. Symmetry at rest was present on average at the time of the second follow-up visit after 5.6 (±1.4) months. All of the patients showed symmetry at rest and a spontaneous smile without the use of the jaw at the time of the fourth follow-up (average 10.8 (±1.8) months).

Needle-EMG examinations were carried out during each examination until a spontaneous smile was obvious. The first reinnervation potentials were detected earliest at the first follow-up visit in three patients (Figure 4). At the second follow-up, after 5.6 (±1.4) months, three patients showed reinnervation patterns without the use of the jaw or biting. At the time of the third follow-up (8.4 (±2.0) months), a spontaneous smile was possible and no EMG was needed. 

### 3.3. Clinical Outcome 

All of the patients demonstrated a spontaneous smile by one year after surgery. Table 3 and Figure 5 and Figure 6 demonstrate the clinical outcome of the patients before surgery and at the one-year follow-up appointment. Both global palsy scores improved significantly after the surgery. However, several additional static procedures were carried out in each of the patients (Table 3).

## 4. Discussion

In the present study, we investigated the time course of reinnervation and the functional outcome after the reanimation of the lower face with a micro-neurovascular anastomosed gracilis muscle flap, supplied by the masseteric nerve. Five patients with longstanding FP, who underwent the surgery between June 2019 and October 2020, were consecutively followed up over the time period of one year. The time course of the gracilis graft reinnervation was observed clinically, and was examined using a needle EMG. 

The major decision points for the reanimation technique were: 1/ one-stage surgery; 2/ the unclear condition of the peripheral facial nerve branches after previous tumor surgery (Patient 1); and 3/ the fiber quality of the masseteric nerve (stronger or wider excursion of the smile). Besides patient 1, all of the patients preferred a single-stage surgery and did not wish to undergo a cross-face reanimation of the gracilis graft. In addition, some patients seemed to prefer the sacrifice of the masseteric function over the facial nerve branches of the healthy side, as is the case with cross-face treatment.

All of the patients showed a preoperative flaccid facial palsy. All of them were able to express a spontaneous emotion-stimulated smile after one year. We noticed a very short reinnervation interval in our patients. The first single reinnervation potentials were detected 3 months after surgery. After 6 months, in three (3/5) patients, clear reinnervation patterns were present in the gracilis graft, which did not require the co-activation of the masseteric muscles (using chewing or biting). Clinically, the patients obtained symmetry of the face at rest after 6 months, and could spontaneously smile without the co-activation of the jaw after an average time of 11 months. In the context of grading, the clinical appearance corresponded to an improvement of two to four points in both the HB and Stennert index. The postoperative outcome was significantly improved in 100% of the cases. This corresponds to the empirical values of other studies [19,20]. Compared to the cross-face reinnervated gracilis graft, in which a good functional outcome can be expected after a period of up to 18 months [8], the patients with a masseteric nerve-supplied gracilis graft achieved the final result considerably earlier. Although all of the patients could express a spontaneous smile in our case series, the co-activation of the jaw and the masseteric muscles is essential to support reinnervation. For this reason, the rehabilitation might be challenging for patients with a lower motivation for training, with a weak muscle tonus, or who are not able to chew or bite (i.e., non-functional dental status). The cross-face transfer, on the other hand, enables the simultaneous control of the muscles due to simultaneous nerve supply, and enables symmetrical mobility [8,19]. Therefore, cross-face supplied gracilis transfer is still considered to be the gold standard for the reanimation of the lower face, especially in young patients [17,21]. Our patients underwent a postoperative physical therapy program to stimulate reinnervation and to increase muscle tonus as soon as the reinnervation pattern was detectable in the EMG. We recommend the use of EMG diagnostics for the early detection of the reinnervation potentials, so that physical therapy is initiated as early as possible. However, some patients might profit from the additional stimulation training of the graft, i.e., electrical stimulation to obtain higher muscle tone and volume [22,23].

Previously published data are inconsistent and ambiguous regarding the ability to develop a spontaneous, emotionally stimulated smile after reconstruction with a free muscle graft innervated via the masseteric nerve. Some studies describe the nerval reinnervation via cross-face transfer as the only option for emotionally stimulated mobility. Bigliolo et al. reported a spontaneous smile after reconstruction via the masseteric nerve in about 10% of the patients [24]. However, there are number of studies which report far better results. For example, Hontanilla et al. described the achievement of spontaneous mobility in 55% of their patients [15], and Manktelow et al. reported success in 60% of their cases [11]. 

Our case series included a somewhat-young patient collective. The good plasticity of the brain might explain the success of the procedure. Another possible explanation for the good outcome might be the close and even overlapping cortical relationship of the responsible cortical areas for smiling and biting. In particular, women and young patients were described to obtain a favorable outcome and good functional results [15].

Compared to the cross-face transfer of the gracilis graft, the masseteric nerve transfer still poses a one-stage surgery and is best suited for patients who wish to obtain a quicker result and a single-surgery procedure. As the gracilis graft only simulates the direction of the zygomatic muscle, we observed a postoperative hollowness of the periorbital region. The patients have to be informed about the possible further treatments of procedures, which might be necessary in the further course of therapy (augmentation, etc.). Furthermore, the paresis of the upper face was not treated here. Therefore, the rehabilitation of the eye region (i.e., lidloading, eyebrow lift, etc.) is often additionally necessary. As mentioned on page 3, four patients underwent supplementary static procedures such as brow lifts, lidloading or lateral canthopexy. The additional procedures for each patient are also named in Table 3, on page 9. These procedures were crucial for the improvement of the ocular closure and overall facial symmetry. 

The dynamic rehabilitation of the eye region is also possible. We prefer the dynamic reanimation of the eye closure (mostly the transposition of the temporal muscle) as a second step of facial reanimation if static procedures do not lead to satisfactory results. Especially in younger patients, static lidloading or brow lifts are easy and quick techniques with good, long-lasting results. In older patients, we would probably recommend dynamic eye rehabilitation, which is slightly more time-consuming, but generally leads to a better long-term result.

Besides the gracilis graft, the common alternatives for muscle replacement are the minor pectoralis, anterior serratus and latissimus dorsi muscles [25].

The gracilis muscle, however, has become the most common muscle graft because of its accessibility, superficial course, and the low morbidity of the sampling site [7,8,11,17,20]. The simultaneous surgical approach of both sites, the face and the thigh, is possible in a two-team approach. 

In our case series, we mostly used the superficial temporal artery and the retromandibular vein plexus as donor vessels for the muscle graft. Despite the small vessel calibers, the blood supply was sufficient in all of the cases. In comparative studies, the facial artery and vein are preferred because of their larger caliber. In 20%, the facial vessels are dissected in the submental triangle. We also used the facial vessel supply in one patient after a radical parotid surgery because the superficial temporal artery was not present. However, in approximately 33% of the patients with preexisting (tumor) surgery, even the facial artery and vein are no longer available, such that alternative blood donors have to be considered [26]. In this case, the submental vessels might be a good donor replacement [27]. Furthermore, it is important to acknowledge that the changed anatomy after radical resection might pose a challenge in the visualization of the anatomical structures. In the case of our patient with a congenital dysplasia of the middle face, the dissection of the masseteric nerve was more difficult but still feasible. 

Faris et al. also described this procedure to be advantageous in patients with FP after radical parotid surgery [9]. Besides the beneficial access to the donor nerve and rapid reinnervation of the graft, the muscle bulk also acts as a kind of filling material after a radical resection of tissue, and improves the symmetry of the face. Furthermore, the fiber quality of the masseteric nerve leads to more powerful reinnervation compared to the cross-face transfer [8].

Postoperative complications occurred in two of our five cases. One patient developed a wound infection of the cheek, which led to a prolonged hospitalization. Another patient underwent a revision surgery of the thigh and neck because of postoperative bleeding which occurred due to an overdose of anticoagulation therapy during his ICU stay. The literature search revealed only few studies which systematically report on complications. Garcia et al. described, in their meta-analysis, an overall complication rate of 9.6%. The majority of the complications were mild, such as wound infections in 3.5 to 5.6%, and hematomas in 3.6% of the cases. Flap failure occurred in 1.8% of cases [20,28].

In summary, dynamic facial reanimation with a free gracilis transfer is a valuable technique to improve oral competence and emotional and nonverbal communication. An improvement in facial symmetry, as well as an increase in quality of life, occured in 100% of the cases with successful muscle transfer, regardless of the choice of donor nerve [19,20]. Hontanilla et al. even discussed whether the treatment with the masseteric nerve should be the preferred alternative when considering all of the advantages, such as the lower donor site morbidity, a more symmetrical and stronger smile, and—in the course of time—also a possible spontaneous mobility [8]. In summary, we consider the micro-neurovascular anastomosed grafting with the gracilis muscle and masseteric nerve to be a sufficient and reliable rehabilitation option for the lower face. Furthermore, considering the increasing data on this topic, we believe that a spontaneous smile is a realistic goal in facial paralysis reanimation with the masseteric nerve. A drawback of our study is the limited number of cases. Therefore, larger numbers of patients are required in order to draw general conclusions.

## 5. Conclusions

In conclusion, micro-neurovascular anastomosed grafting using the gracilis muscle flap is a sufficient and reliable rehabilitation option for the lower face. The nerve supply via the masseteric nerve allows the very rapid and strong reinnervation of the face, and results in a spontaneous smile within 10 months of the surgery.

## Figures and Tables

**Figure 1 diagnostics-12-01282-f001:**
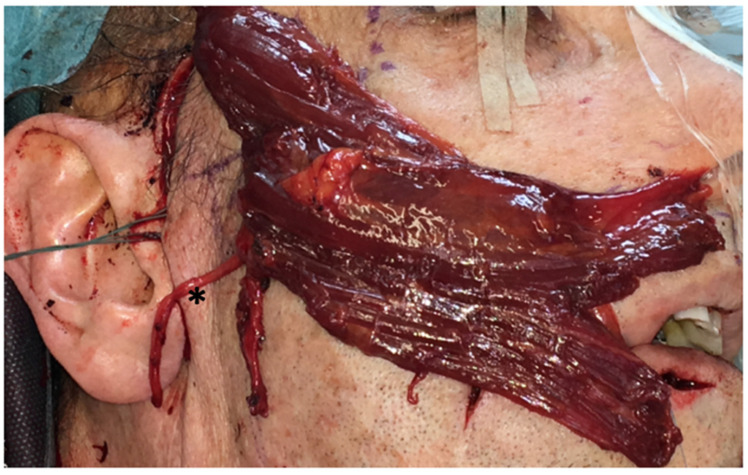
The gracilis muscle graft before implantation in the planned orientation on the right side. * marks the obturator nerve.

**Figure 2 diagnostics-12-01282-f002:**
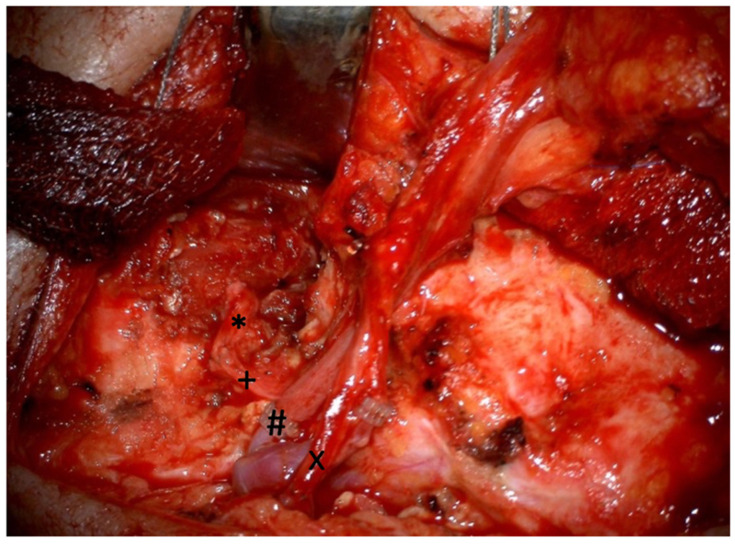
Operative situs on the right side after the completion of the nerve and vessel sutures using the superficial temporal artery and retromandibular vein. * marks the masseteric nerve, + marks the obturator nerve, # marks venous anastomosis, and x marks the arterial anastomosis.

**Figure 3 diagnostics-12-01282-f003:**
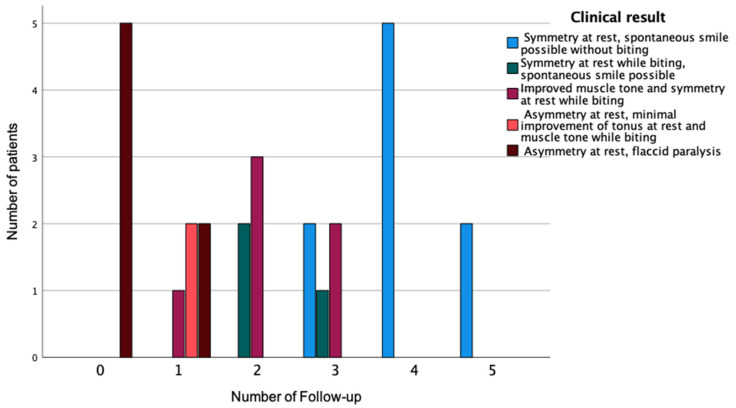
Functional outcome at the first to fifth follow-up visit. 0 = preoperative state, 1–5 = number of the follow-up visit.

**Figure 4 diagnostics-12-01282-f004:**
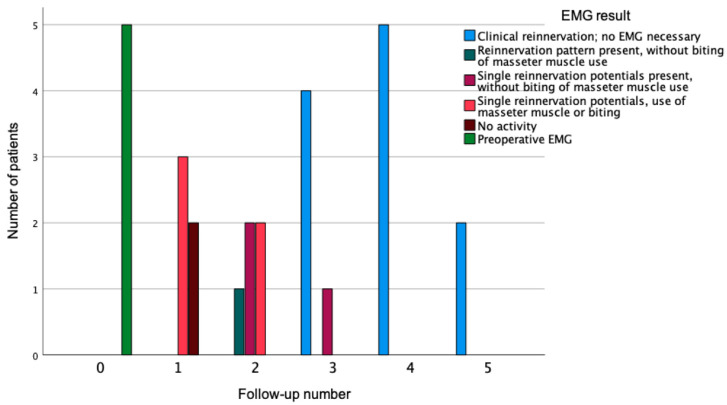
EMG-result at the first to fifth follow-up visit. 0 = preoperative state, 1–5 = number of the follow-up visit.

**Figure 5 diagnostics-12-01282-f005:**
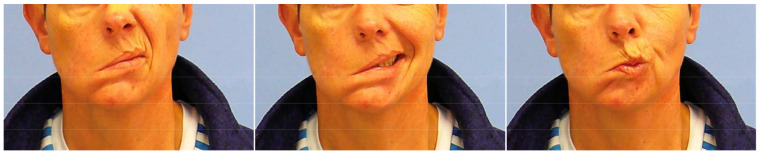
Preoperative excursion of the mouth in patient 4. **Left**: wrinkling the nose; **center**: smiling; **right**: pointing the mouth.

**Figure 6 diagnostics-12-01282-f006:**
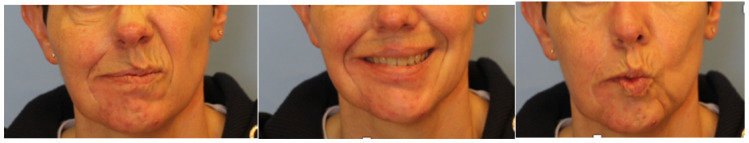
Postoperative result 16 months after the procedure in patient 4. **Left**: wrinkling the nose; **center**: smiling; **right**: pointing the mouth.

**Table 1 diagnostics-12-01282-t001:** Patients’ characteristics.

Patient	Gender	Age in Years	Duration of Palsy in Years	Etiology	Preoperative Facial Grading (House Brackmann (HB) Score)
1	male	68.1	2.0	Radical parotidectomy and postoperative Radiation, Parotid Carcinoma	VI
2	female	48.6	10.5	Idiopatic	VI
3	male	48.0	23.1	Vestibular Schwannoma	VI
4	female	23.8	4.1	Facial Nerve Schwannoma	IV
5	female	55.0	55.1	Congenital dysplasia of the middle and lower face	V

**Table 2 diagnostics-12-01282-t002:** Follow-up timeline in months, average.

Follow-Up Visit	Time from Surgery, Months (±SD)	Time to Previous Visit, Months (±SD)
1	3.1 (1.0)	3.2 (±1.0)
2	5.6 (1.4)	2.5 (±0.6)
3	8.4 (2.0)	2.8 (±0.9)
4	10.8 (1.8)	3.1 (±0.7)
5	16.1 (0.4)	4.9 (±2.5)

**Table 3 diagnostics-12-01282-t003:** Facial paralysis scores, preoperatively and at the last (one-year) follow-up appointment.

Patient	Timepoint of Documentation	House Brackmann Score	Stennert Index	Additional Surgical Procedures
1	PreoperativeLast follow-up	6	4/6	Forehead-/Browlift, Lidloading
	2	0/3	
2	PreoperativeLast follow-up	6	4/6	Forehead-/Browlift
	2	1/1	
3	PreoperativeLast follow-up	6	4/6	Canthopexy, Lidloading
	3	1/4	
4	PreoperativeLast follow-up	42	3/51/1	Canthopexy, Lidloading
5	PreoperativeLast follow-up	5	3/4	-
	3	2/1

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
