# Peer review of "Reanimation of the Smile with Neuro-Vascular Anastomosed Gracilis Muscle: A Case Series"

_diagnostics, 2022, doi:10.3390/diagnostics12051282_

Round 1

Reviewer 1 Report

This is an interesting manuscript, focusing ingredients on dynamic facial reanimation with gracilis muscle reinnervated with masseteric nerve.

This manuscript focuses on a series of long-standing facial paralysis cases (more than 10 years in 3/5 cases), and shows the following intrinsic limitations:

  • the series is small (5 cases) because of a narrow enrollment period (just 15 months)
  • the etiology of facial palsy was quite heterogeneous (including post-surgical, idiopathic, and congenital cases).

However, Authors should be apprised for their effort to share their results, which are potentially beneficial, given the substantial lack in literature of large series on gracilis flap re-innervated with masseteric nerve.

Few points should be better addressed in the discussion:

  • one of the things which stand out from this paper is the fact that all patients achieved a spontaneous smile after treatment. Biglioli et al. (2012, JOMS) state about the same procedure that "the use of the masseteric nerve in facial reanimation procedures does not achieve spontaneous smiling function". Authors should discuss the possible reasons of such an outcome difference compared to the previous reports of the same technique.
  • Most patients underwent static procedures on the upper part of their face. What is the contribution to the overall facial symmetry of dynamic and static procedures in this series of patients?

Regarding presentation of clinical results, I'd suggest, for clarity sake, to use an alluvial plot instead of the histogram in fig. 3, to show the clinical trajectory of each patients at different follow-up time-points.

English should be edited for style.

Reviewer 2 Report

Manuscript is nicely written. I have no suggestions to improve it.

Reviewer 3 Report

" Reanimation of the Smile with Neuro-Vascular Anastomosed Gracilis Muscle "

It is very interesting to describe the surgical procedure and the time- course of clinical and electromyographical reinnervation after reanimation of the smile using the gracilis graft, which is reinnervated with the masseteric nerve. However, there are a few corrections that are essential to meet the standard for publication. Please refer to the following comments.

1) Please specify that the title is a case series.

2) Please describe the criteria for selecting this procedure.

3) There are only 5 cases in your case series. Therefore, it is necessary to explain all cases. Please explain the changes before and after the simple surgery and explain each process using figures. It is especially important to visually show the changes before and after surgery.

4) Please add the limitations and prospects of this study. Please add your opinion to the discussion section.

Round 2

Reviewer 3 Report

Thank you for giving me this opportunity to re-review your revised manuscript.

I am happy that all of the suggested corrections have been made.

Thank you for spending so much time for revised manuscript.